# Hydrogen Sulfide and Irisin, Potential Allies in Ensuring Cardiovascular Health

**DOI:** 10.3390/antiox13050543

**Published:** 2024-04-28

**Authors:** Lorenzo Flori, Giada Benedetti, Vincenzo Calderone, Lara Testai

**Affiliations:** 1Department of Pharmacy, University of Pisa, Via Bonanno 6, 56120 Pisa, Italy; lorenzo.flori@farm.unipi.it (L.F.); giada.benedetti@phd.unipi.it (G.B.); vincenzo.calderone@unipi.it (V.C.); 2Interdepartmental Research Center Nutrafood “Nutraceuticals and Food for Health”, University of Pisa, 56120 Pisa, Italy; 3Interdepartmental Research Centre of Ageing Biology and Pathology, University of Pisa, 56120 Pisa, Italy

**Keywords:** gasotransmitter, myokine, cardiovascular risk, crosstalk, benefits

## Abstract

Irisin is a myokine secreted under the influence of physical activity and exposure to low temperatures and through different exogenous stimuli by the cleavage of its precursor, fibronectin type III domain-containing protein 5 (FNDC5). It is mainly known for maintaining of metabolic homeostasis, promoting the browning of white adipose tissue, the thermogenesis process, and glucose homeostasis. Growing experimental evidence suggests the possible central role of irisin in the regulation of cardiometabolic pathophysiological processes. On the other side, hydrogen sulfide (H_2_S) is well recognized as a pleiotropic gasotransmitter that regulates several homeostatic balances and physiological functions and takes part in the pathogenesis of cardiometabolic diseases. Through the S-persulfidation of cysteine protein residues, H_2_S is capable of interacting with crucial signaling pathways, exerting beneficial effects in regulating glucose and lipid homeostasis as well. H_2_S and irisin seem to be intertwined; indeed, recently, H_2_S was found to regulate irisin secretion by activating the peroxisome proliferator-activated receptor gamma coactivator 1-alpha (PGC-1α)/FNDC5/irisin signaling pathway, and they share several mechanisms of action. Their involvement in metabolic diseases is confirmed by the detection of their lower circulating levels in obese and diabetic subjects. Along with the importance of metabolic disorders, these modulators exert favorable effects against cardiovascular diseases, preventing incidents of hypertension, atherosclerosis, heart failure, myocardial infarction, and ischemia–reperfusion injury. This review, for the first time, aims to explore the role of H_2_S and irisin and their possible crosstalk in cardiovascular diseases, pointing out the main effects exerted through the common molecular pathways involved.

## 1. Introduction

The increasing incidence of obesity in the population requires attention regarding its serious complications. It turns out that, in addition to typical, well-known metabolic complications, obesity, as a systemic disease, carries the risk of non-metabolic complications, among which are cardiovascular (CV) diseases. Currently, they are deemed to be the main reason for death, including stroke, ischemic cardiomyopathy, and atherosclerosis [1].

H_2_S is a pleiotropic gasotransmitter that regulates several homeostatic balances and physiological functions; furthermore, it has also been implicated in the pathogenesis of CV and non-CV diseases [2,3,4]. Very recently, the role of H_2_S in metabolism has emerged, and its contribution to the regulation of lipidic management has been suggested. In this regard, H_2_S is often labeled as a new adipokine, and numerous researchers are exploring the cardiovascular implications consequent to the metabolic influences of this gasotransmitter [5]. On the other hand, irisin is a myokine secreted by several tissues under the influence of physical activity and exposure to low temperatures and through different exogenous stimuli, and it is deeply involved in the maintenance of metabolic homeostasis, promoting the browning of white adipose tissue, the thermogenesis process, and glucose homeostasis [6]. However, growing evidence suggests the possible central role of irisin in the regulation of the CV system [7].

Interestingly, the H_2_S and irisin systems appear to be intertwined. The present unique paper demonstrates the close correlation between these two mediators. Specifically, H_2_S promotes, through the S-persulfidation of PGC1α, the release of irisin in skeletal muscle [8]. Therefore, this review focuses on exploring the intracellular targets underlying their multiple beneficial effects and brings to light possible common aspects, which suggest the existence of crosstalk at the CV level.

## 2. Role of Hydrogen Sulfide in Cardiovascular System

### 2.1. Biosynthesis of Hydrogen Sulfide

In mammalian tissue, H_2_S is biosynthesized in enzymatic and non-enzymatic ways. The non-enzymatic pathway represents the least important one and consists of the reduction of elemental sulfur to H_2_S with the oxidation of six molecules of glucose. Concerning the enzymatic pathway, aminoacid L-cysteine may be the substrate of three enzymes: cystathionine γ-lyase (CSE), cystathionine β-synthase (CBS), and 3-mercaptopyruvate sulfurtransferase (3-MST). In particular, CBS is predominant in the central nervous system (CNS), while CSE is mainly involved at the cardiovascular (CV) level. More recently, the mitochondrial enzyme 3-MST was identified. 3-MST contributes to the endogenous production of H_2_S, cooperating with the cysteine aminotransferase (CAT) [9].

H_2_S modulates a plethora of target proteins through the direct post-translational modification of sulfhydryl groups of cysteine residues; this process is called S-persulfidation. On the basis of this mechanism, H_2_S has been implicated in several pathways at the CV and non-CV levels, demonstrating the deep involvement of this gasotransmitter in the regulation of different functions [2].

Regarding the catabolism of H_2_S, it may follow different pathways: being a reducing agent, it can be consumed by several oxidant factors present in many tissues; moreover, H_2_S can be oxidated at the mitochondrial level, where it is rapidly converted into sulfate and sulfite species through the sequential action of quinone oxidoreductase, rhodanese, and sulfur dioxygenase [10].

### 2.2. Role of Hydrogen Sulfide in Cardiovascular Disorders

H_2_S is deeply involved in the homeostasis of the CV system. In this regard, CSE mRNA was first discovered in vascular smooth muscle cells and subsequently also found in the vascular endothelium, where H_2_S is biosynthesized and then spreads as an endothelial-derived relaxing factor. In fact, the physiological levels of H_2_S, maintained in part by the proper functionality of endogenous CSE, promote the maintenance of the structural and functional integrity of vascular tissues through the engagement of multiple targets, including the stimulation of potassium channels. To the best of our knowledge, H_2_S was first identified as an activator of ATP-sensitive potassium (K_ATP_) channels and, subsequently, voltage-gated potassium (Kv7.4) channels [11,12]. This action is correlated with the hyperpolarization of the cell membrane and then with vasorelaxation. However, other mechanisms contribute to the vasoactive effects of H_2_S; indeed, this gasotransmitter has been proven to inhibit the PDE enzyme [13], and furthermore, it engages in close crosstalk with NO. Interestingly, H_2_S causes a concentration-dependent relaxation of aortic rings, increasing the cyclic guanosine monophosphate (cGMP) level; on the other hand, the inhibition of endothelial NO-synthase (eNOS) or endothelium removal markedly attenuates H_2_S-induced vasorelaxation in rat aortic tissues [14]. Furthermore, H_2_S is also endogenously produced in cardiomyocytes, ensuring protective effects through several mechanisms, among them, the stimulation of mitochondrial potassium (mitoK) channels. They represent a crucial target in cell protection against several types of injuries, including ischemia–reperfusion damage. Abundant evidence points out that, like vascular channels, H_2_S may promote the opening of mitoK_ATP_ and, more recently, mitoKv7.4 channels [15,16]. The activation of mitoK channels promotes the moderate depolarization of the mitochondrial membrane (by engaging certain exchangers, such as the K^+^/H^+^ antiporter, to stop a potentially aberrant depolarization), which reduces the driving force for calcium entry into the matrix and then reduces the probability of the assemblage of mitochondrial permeability transition pores (MPTPs), thus preserving the integrity of these organelles [17]. However, other intracellular pathways have been described in relation to the beneficial effects of H_2_S on the heart, among which are the containment of inflammation and reduction in oxidative stress. In fact, in mice submitted to LPS-induced injury, H_2_S reduces the inflammasome NLRP3 and subsequently activates caspase-1 [18]; a H_2_S treatment can attenuate tissue levels of pro-inflammatory cytokines and activate the nuclear translocation of antioxidant transcription factor Nrf2 in a pre-clinic model of ischemic insult [19].

It is well known that several CV disorders are associated with a deficiency in the endogenous production of H_2_S. Indeed, changes in H_2_S signaling, including its synthesizing enzymes, can lead to CV diseases. Conversely, H_2_S-based interventions have been proven to be beneficial in the prevention of CV diseases, assuming the protective effects of H_2_S at the onset of CV diseases.

In CSE-KO mice, endothelial dysfunction, with consequent hypertension and atherosclerosis, has been observed [14,20,21,22,23,24,25]. These values are comparable to eNOS-KO mice, and endothelium-mediated vasorelaxing activity was reduced by approximately 60% [26]. Even the administration of the CSE inhibitor D,L-propargylglycine (PAG) is associated with an increase in blood pressure in rats [27]. The deficient functionality of the CSE/H_2_S system leads to the development of vascular remodeling and atherosclerosis through increased aortic intimal proliferation and the increased expression of adhesion molecules in CSE-KO mice subjected to an atherogenic diet [25].

Moreover, in one study, CSE-KO mice showed significantly worsened pulmonary edema and cardiac enlargement 12 weeks after a transverse aortic constriction procedure compared with wild-type mice. In particular, cardiac dysfunction progressively worsened from 3 to 12 weeks after transverse aortic constriction. This pattern was associated with the significant dilation of the left ventricle compared with the wild-type group [28].

Experimental evidence underlines the importance of endogenous H_2_S production in the vascular endothelium, demonstrating that both the pharmacological inhibition of CSE and CSE knockdown show a reduction regarding vascular endothelial growth factor (VEGF)-induced angiogenesis in mouse aortic rings, the impaired processes of migration and germination of endothelial cells, and the loss of endothelium-dependent acetylcholine-induced vasorelaxation [26,29].

Abundant experimental evidence has described a reduction in H_2_S levels in preclinical models of obesity. Rats fed with a high-fat diet (HFD) have shown a condition of oxidative and inflammatory stress that is able to reduce both the expression and the activity of vascular CSE, with a consequent decrease in H_2_S production [25,30].

The correlation between H_2_S levels, the expression of the enzymes involved in its endogenous production, and diabetes outlines an even more complicated scenario. A reduction in circulating levels of H_2_S has been described in non-obese diabetic mice [31], in rats with streptozotocin-induced diabetes [32,33], and in patients with type 2 diabetes (T2D) [32,34]. However, no deficits in the expression of CSE, CBS, and 3-MST have been found either in endothelial cells subjected to high glucose concentrations or in the thoracic aortas of rats with streptozotocin-induced diabetes [32,35]. More recent data have focused on the possible inhibition of endothelial 3-MST and a simultaneous increase in H_2_S consumption by endothelial cells when subjected to high glucose concentrations [32,35].

Although, from a preclinical point of view, the role of H_2_S in CV diseases has been widely evaluated, evidence relating to plasma levels in humans, both in physiological and pathological conditions, and its exogenous administration as a pharmacological strategy is still lacking.

Plasma levels of H_2_S are significantly lower in patients with coronary heart disease than normal control individuals, as well as in patients with unstable angina and acute myocardial infarction (MI) compared with healthy age-matched control individuals [36]. The same trend is present in patients with heart failure, in whom circulating H_2_S levels are markedly lower compared with same-age healthy controls [37].

A detailed meta-analysis analyzed 21 studies and confirmed a general reduction in plasma H_2_S levels in elderly subjects suffering from age-associated pathologies, such as CV diseases, diabetes, and cancer. A stratification of the results obtained revealed an interesting difference between plasma H_2_S levels in pathologies characterized by high acute inflammatory conditions and pathological scenarios characterized by persistent oxidative stimuli and chronic sub-clinical inflammatory processes. While, in the first case, there was a generalized increase in circulating H_2_S levels, in the chronic condition of meta-inflammation, the plasma H_2_S levels were deficient. The results analyzed by Piragine and colleagues, therefore, confirmed that plasma H_2_S levels were reduced in elderly subjects suffering from CV diseases [38]. The analysis of the role of this gasotransmitter in lipid metabolism is very recent; however, although preliminary, challenging hypotheses have emerged [5]. In fact, some recent works have gathered experimental evidence demonstrating how H_2_S can modulate adipogenesis through peroxisome proliferator-activated receptor γ (PPARγ), modulate lipolysis by inhibiting the phosphorylation of perilipin-1 (plin-1), and finally promote the browning process through the release of irisin from skeletal muscle [8,39,40].

We can, therefore, state, in relation to the large amount of supporting experimental evidence, that alterations in circulating H_2_S levels are observed from the early stages of the onset of CV diseases [41]. This paves the way to start looking at serum H_2_S levels as a possible diagnostic marker of the onset of CV diseases (Figure 1). In this regard, however, we must face a non-negligible problem in the sensitivity and specificity of the analytical methods currently available for the evaluation of circulating H_2_S levels. Furthermore, this scenario is further complicated by the complexity of the physiological homeostasis of H_2_S, its binding to plasma proteins, and the variability of its balance with undissociated forms. Regarding physiological pH, approximately two-thirds of H_2_S exists as its first dissociation product (hydrosulfide ion, HS^−^); the remaining part is present in an undissociated form (H_2_S), while its second dissociation product (sulfide ion, S^2−^) is present in negligible quantities [37,42,43].

The underlying issues for the detection of circulating H_2_S levels make it difficult to define an accurate range. The levels of circulating free sulfides are in the order of low nM, but there are reserves of bound sulfides in the blood that can be released in several pathophysiological conditions. These reserves serve as a feedback mechanism of free sulfide levels and both circulating and tissue sulfide consumption. Furthermore, exogenously administered sulfide species may be partly bound to blood cells [44].

## 3. Role of Irisin in Cardiovascular System

### 3.1. Biosynthesis of Irisin

In 2012, Boström and colleagues first discovered irisin [45], a myokine commonly present in several districts of the human body, including the heart, the brain, skeletal muscle, the liver, the kidneys, the lungs, the spleen, skin, and adipose tissue; however, its levels vary on the basis of tissue distribution. This myokine derives from the proteolytic cleavage of FNDC5, a protein located in the cytoplasm of the C-terminal fragment, while the N-terminal portion leans out in the extracellular space [6]. It can be present as either a homodimer or a dimer formed by a β-sheet between the Arg75 and Glu79 amino acids, which, in turn, stabilizes the structure. Moreover, irisin is glycosylated before being secreted, and such post-translational modification preserves its biological and functional capacity [46]. Finally, the integrin complex, specifically, αV/β5 integrin, seems to be the specific receptor through which irisin affects biological targets [47].

PGC-1α is a main trigger of irisin release; indeed, PGC-1α mutations decrease FNDC5 expression, while the overexpression of PGC-1α, complexed with CREB, increases the cleavage of FNDC5 in C2C12 cells [48]. However, other regulators have emerged, including AMP-activated protein kinase (AMPK) [49]. In this regard, the treatment of obese mice with icariin is associated with a significant reduction in body weight gain, the increased expression of FNDC5 protein, and the phosphorylation of AMPK; conversely, the use of the AMPK antagonist, compound C, or the silencing of AMPK, deletes these effects [49].

### 3.2. Role of Irisin in Cardiovascular Disorders

It is known that irisin is released into the blood from skeletal muscle cells under the influence of exercise or exposure to cold. Interestingly, high-intensity exercise results in a greater release of irisin compared with low-intensity exercise [50]. In this regard, a widely investigated tissue target is white adipose tissue owing to the recognized capability of irisin to promote the browning process and contribute to its “healthy” modulation. Indeed, with the browning process, white adipocytes acquire characteristics similar to brown adipocytes: increased expression of uncoupled protein 1 (UCP1), improved metabolic activity, lipid accumulation, and reduction in cell size (for this reason, usually, these cells are called beige adipocytes) [5]. Furthermore, growing evidence suggests that irisin may also play a central role in other districts of the organism, including the CV system [7]. For example, on one side, experimental evidence from histological analyses has demonstrated that the exogenous administration of irisin influences vascular morphology in rats, increasing the thickness of the intima–media complex and stimulating vascular remodeling at the aortic level, increasing the number of smooth muscle cell nuclei and the number of elastic lamellae in the medial layer [51,52]. On the other side, irisin has been shown to stimulate angiogenesis in both in vitro and in vivo models, promoting the migration and formation of new capillaries [53]. Such processes are considered crucial in the progression of CV morbidities; in fact, neovascularization and angiogenesis may contribute to restoring blood supply and minimizing myocardial damage.

Moreover, a recent study highlighted the vasorelaxant effect of irisin on endothelium-intact and -denuded rat aortic rings. In particular, this study was the first to report that irisin-induced relaxation responses are associated with the stimulation of potassium channels in the specific K_V_, K_ATP_, and calcium-activated potassium (KCa) channels, suggesting an active effect on vascular tone, in addition to anti-atherosclerosis and pro-angiogenic effects [54].

Despite the efforts of various researchers, the role of irisin in the myocardium is still controversial; on the one hand, acute MI patients with elevated serum irisin levels are associated with a higher rate of adverse CV events, suggesting that an excess of this myokine could induce mitochondrial dysfunction and increase cardiomyocyte damage [55,56]. On the other hand, the treatment of isolated hearts with irisin markedly reduces the infarct size [57], lowers lactate dehydrogenase levels, and suppresses mitochondrial swelling [58], indicating that the irisin treatment may be viewed as a novel approach to eliciting cardioprotection [59]. Of note, the irisin treatment suppresses high oxidative stress and the relative concentrations of MDA caused by hypoxic injury, and recently, it was proven to reduce ferrous iron levels, indicating an important antiferroptotic role [60]. Other authors have reported that irisin is able to prevent Ang II-induced hypertrophy in cardiomyocytes [61]; accordingly, hypertrophic processes and alterations have been described in cardiomyocytes following the silencing of the gene coding for FNDC5. Irisin incubation reduces the expression of genes involved in fibrotic and hypertrophic processes, such as TGFβ1, fibronectin, and collagen I [62]. The research on putative mechanisms underlying the effects of irisin on the heart is still embryonic and inconclusive; however, the stimulation of some kinases, including the AMPK pathway, and the attenuation of inflammasome via NLRP3 have been supposed [7].

Interestingly, circulating irisin levels are inversely correlated with the incidence of coronary artery calcification progression, and more generally, they are directly correlated with CV risk factors. A meta-analysis of seven case–control studies on a total of 867 patients and 700 controls demonstrated that irisin concentrations were significantly lower in patients with coronary artery disease compared with healthy controls. On this basis, serum irisin levels can be considered a potential biomarker of the presence of coronary artery disease and its severity [63,64,65,66].

Patients with chronic CV diseases show stable, albeit lower, circulating levels of irisin compared with healthy subjects. On the other hand, serum irisin concentrations gradually decrease over 48 h after acute MI. Additionally, a clinical study identified reduced circulating levels of irisin in patients with MI and heart failure compared with those with MI but without heart failure [67,68]. In contrast, further experimental evidence has identified elevated levels of irisin 28 days after ST-elevation MI, and these were associated with an increased risk of adverse CV events 3 years after the event, such as recurrent MI and episodes of angina, ischemic stroke, new or worsening episode of congestive heart failure, and coronary revascularization procedures [69].

Notably, Remuzgo-Martínez and colleagues evaluated the degree of subclinical atherosclerosis in a large cohort of 725 patients with axial spondyloarthritis, identifying how low levels of irisin are associated with a high atherogenic index (AI > 4), calculated as a ratio between total cholesterol and high-density lipoprotein cholesterol (HDL-C) and carotid plaques [70]. Indeed, patients suffering from axial spondyloarthritis, in addition to the impact of the rheumatic disease itself, are subject to the accelerated development of atherosclerosis and, therefore, to a greater incidence of CV diseases.

In general, irisin serum levels are also affected by obesity, diabetes, and lipid metabolism alteration, revealing a negative correlation between irisin and total cholesterol (TC), triglyceride (TG), HDL-C, low-density lipoprotein cholesterol (LDL-C), and serum homocysteine (Hcy) [46,71,72,73]. Accordingly, many studies have shown that irisin has a potential role in metabolic diseases, such as diabetes and obesity, which can probably be linked to evidence of the regulation of metabolic peptides such as insulin, glucagon, and leptin. In one study, 65 patients with T2D showed serum irisin levels inversely correlated with body mass index, hyperglycemia, and body fat percentage, suggesting that this myokine might represent a predictive marker for the onset and progression of CV diseases. On the other hand, although these patients had reduced flow-mediated dilation, no significant correlation was observed with serum irisin levels compared with healthy subjects [74].

Dong and colleagues brought to light the correlation between low irisin levels and CV and cerebrovascular diseases with a retrospective cohort study of 152 enrolled hemodialysis patients. Among 55 deaths, 18 attributed to CV and cerebrovascular events had significantly lower serum irisin levels than the 97 patients in the survival group [75].

Serum irisin levels were directly correlated with the functional outcomes of Chinese patients with acute ischemic stroke (AIS) in a 3-month follow-up study. The results demonstrated that low levels of irisin were associated with unfavorable early functional outcomes in patients with AIS [76].

Finally, the correlation between systemic irisin levels and hypertension appears to be rather complex. Increased levels of irisin are associated with hypertension both in adults and in overweight children with high levels of diastolic and systolic blood pressure [77,78]. A cross-sectional study including 532 patients with chronic kidney disease indicated a correlation between high irisin levels and increased diastolic blood pressure [79]. In contrast, reduced diastolic and systolic blood pressure is associated with reduced circulating levels of irisin in women with preeclampsia [80].

The data collected in recent years indicate that this myokine is a potential diagnostic biomarker of CV diseases and that it is also useful in the stratification of patients in relation to characteristic comorbidities, such as obesity, T2D, and insulin resistance. Furthermore, it has been shown that different levels of systemic irisin can contribute to the assessment of CV disease severity. Irisin also appears to be a promising preventive biomarker in MI, suggesting that irisin concentrations may be related to the onset of an acute ischemic event. Finally, patients with higher circulating levels of irisin are associated with a higher survival rate, allowing this myokine to be “crowned” as a potential positive prognostic biomarker (Figure 1, Table 1).

## 4. Crosstalk between Irisin and Hydrogen Sulfide

A wide body of evidence suggests that irisin and H_2_S are regulators of a plethora of targets involved in CV homeostasis. However, to the best of our knowledge, only one study has investigated the hypothesis that H_2_S could be considered a regulator of irisin biosynthesis. Parsanathan and colleagues observed that the modulation of H_2_S synthesis, using CSE inhibitors, CSE siRNA, or H_2_S donors, affected not only PGC1α and FNDC5 but even irisin expression. This important study showed that irisin circulating levels are correlated with H_2_S ones. In vivo studies on HFD-fed mice presenting obesity and diabetes conditions have demonstrated a reduction in H_2_S levels, along with decreased levels of irisin and FNDC5 expression in mouse muscles. Similar results have been observed in in vitro studies on myotubes exposed to high glucose or palmitate (which imitate diabetic conditions), where lower levels of CSE, PGC-1α, and FNDC5 expression have been reported. Moreover, CSE-KO myotubes or myotubes treated with CSE inhibitors, such as propargylglycine and aminooxyacetate, impair not only glucose uptake but also FNDC5, PGC-1α, and irisin expression. The importance of the H_2_S system in irisin release has been demonstrated even in a cell treatment with an H_2_S donor, which upregulated irisin and glucose uptake. Thus, the underlying supposed mechanism is the H_2_S activation of PGC-1α, which upregulates irisin and favors glucose uptake. Nevertheless, a clear demonstration of the mechanisms underlying this crosstalk is missing [8]. Further studies are necessary to clarify the relationship between these endogenous modulators, but it is well established that they share several intracellular pathways that are the basis of their beneficial effects and are involved in CV and metabolic disorders. Since these two important mediators act on crucial proteins that belong to intertwined pathways, an interesting form of crosstalk between H_2_S and this myokine might be assumed. In the following paragraphs, the main H_2_S- and irisin-modulated pathways are analyzed, focusing on possible points of synergism (Table 2).

### 4.1. Effects of H_2_S and Irisin on the SIRT1/AMPK/PGC1α Pathway

SIRT1/AMPK/PGC1α cascade is a well-recognized target for H_2_S, and now, the S-persulfidated cysteine residues of SIRT1 (Cys371, Cys374, Cys395 and Cys398) have been identified [2,83,127]. Considering the central role played by SIRT1 in cellular metabolism during aging—as well as in the progression of chronic age-related pathologies and considering the evidence of the contribution of H_2_S to its positive modulation—interest in this gasotransmitter has undoubtedly grown. Some decisive studies have brought to light the modulation of SIRT1 by means of the CSE/H_2_S system. Wu et al. found an increment of SIRT1 activity and expression in NaHS-exposed cardiomyocytes [81]. In a murine model of ischemia–reperfusion (I/R) injury, Hu et al. observed that NaHS showed a protective effect, reducing the size of infarcts, lowering myocardial enzymes, and restoring cardiac functions; conversely, the inhibition of SIRT1 by the selective inhibitor EX-527 prevented protective activity [82]. Finally, in a model of heart failure induced by isoproterenol treatment, NaHS (56 µmol/kg/day) increased cardiac hypertrophy markers (ANP and BNP) after 4 weeks, improved cardiac function, and decreased myocardial fibrosis. Moreover, the downregulated expression of SIRT1 and PGC1α was significantly restored by NaHS treatment. Specifically, H_2_S directly increased the activity of SIRT1 via S-persulfidation at its zinc figure domains and reduced the level of acetylated PGC1α [83].

Shimizu et al. performed in vivo experiments to study the role of H_2_S as a regulator of mitochondrial biogenesis via the AMPK/ PGC1α pathway. In CSE TG^+^ mice, the mitochondrial content in the heart was higher than in CSE KO mice; an increase in these levels was even observed in mice treated with SG1002 (20 mg/kg/day), a well-known H_2_S donor. Moreover, unlike in CSE KO mice, the PGC1α content shifted to its nuclear expression, reducing the cytosolic expression in CSE TG^+^ mice. This result suggested that both endogenous and exogenous H_2_S affect mitochondrial heart content and, in particular, the transcriptional activity of PGC1α [84].

Treatment with SG1002 enhances SIRT1 activity and AMPK phosphorylation; in contrast, AMPK-deficient mice treated with SG1002 (20 mg/kg/day) for 4 weeks do not show changes in PGC1α expression at the cardiac level, suggesting that H_2_S regulates mitochondrial biogenesis through the SIRT1/AMPK/PGC1α axis [84].

The SIRT1/AMPK/PGC1α cascade has also been described for irisin. Indeed, PGC1α is the main modulator of the expression of the irisin precursor FNDC5 [45]. Since H_2_S regulates SIRT1 and PGC1α activity, two of the multiple factors involved in irisin biosynthesis, an interesting indirect modulation of irisin synthesis has been suggested [8]. Therefore, like H_2_S, irisin protects damaged cardiomyocytes via a SIRT1-mediated mechanism; so, the shared PGC1α/SIRT1 pathway between irisin and H_2_S suggests a synergic mechanism that enhances cardioprotective action.

Furthermore, several pharmacological (metformin) [128] and nutraceutical treatments (including resveratrol and ginestein) [129,130] can induce the expression of AMPK/SIRT1/PGC1α and raise irisin levels.

### 4.2. Effects of H_2_S and Irisin on the eNOS/NO System

Endothelium dysfunction is an early phase in the pathogenesis of atherosclerosis and is correlated with CV diseases [131,132]. Since impaired H_2_S synthesis is associated with the pathogenesis of endothelial dysfunction, crosstalk between NO and H_2_S has been widely investigated, and several findings suggest that H_2_S can be considered a supporter of NO in the regulation of vascular tone [14,133]. In this regard, Jin et al. observed that NaHS in L-NAME-induced hypertensive rats increased the plasmatic concentration of H_2_S and promoted the Akt/eNOS/NO pathway, preventing L-NAME-induced hypertension; moreover, it attenuated remodeling and cardiac dysfunction [89]. Similar results were obtained in another study, in which isolated hearts from frogs and rats were perfused with NaHS [86]. King et al. observed that CSE KO mice showed a significant reduction in phosphorylation at the eNOS activation site and an increased myocardial infarct size after I/R injuries. Acute H_2_S treatment restored eNOS function and NO bioavailability and attenuated I/R injuries. In contrast, in eNOS KO mice, H_2_S was unable to reverse I/R injuries or eNOS phosphorylation, suggesting that H_2_S’s cytoprotective effects are mediated by an eNOS/NO-dependent mechanism [90]. Similar results have been obtained by other authors [87,88,134].

Concerning irisin, Han et al. performed both in vivo and in vitro studies to investigate whether it is directly involved in the regulation of endothelium homeostasis in hypercaloric diet-induced obese mice. C57/BL6 mice fed with HFD and treated with an intraperitoneal injection of irisin (0.5 µg/g/day) for 8 weeks showed enhanced Ach-induced vasorelaxation and improved endothelial function compared with untreated obese mice. This effect was abolished when aortic rings were incubated with the NO synthase inhibitor, L-NAME, and partly attenuated in the presence of the Akt and AMPK inhibitors. Moreover, using the Greiss reaction method, increased NO production was observed in the treated group compared with untreated obese mice, and an increment of phosphorylated AMPK, Akt, and eNOS was measured. Finally, human umbilical vein endothelial cells incubated with irisin (50 nM) showed a significant increment of phosphorylated AMPK, Akt, and eNOS in a time-dependent manner, while pretreatment with the AMPK inhibitor abolished the irisin’s activity [91].

Interestingly, Fu et al. reported the anti-hypertensive effects of irisin (at doses of 0.1, 1, and 10 µg/kg i.p.) in spontaneously hypertensive rats. Mesenteric arterial segments preincubated in the presence of irisin (3000 ng/mL) partly prevented vasoconstriction induced by phenylephrine (1 nmol/L to 10 µmol/L). Therefore, an increment of NO production was observed in endothelial cells, and it was partially blocked by L-NAME [92]. In vitro studies on endothelial cells have shown that irisin incubation increases eNOS, AMPK, and Akt phosphorylation in a concentration-dependent manner, and pretreatment with an AMPK inhibitor blocks this phosphorylation. These findings suggested the direct influence of irisin on endothelial dysfunction via the direct stimulation of NO production [92]; although the role of irisin in endothelial damage remains unclear, this evidence demonstrates the importance of this myokine in regulating endothelial homeostasis, and it is considered an ally in the prevention and treatment of endothelial dysfunction. Lu et al., studying the role of irisin in diabetes-induced endothelial damage and atherosclerosis, observed that treatment with irisin (2 µg/mouse, twice a week) improved Ach-induced vasorelaxant activity. Finally, no protective effect has been observed in the presence of the Akt, AMPK, or eNOS inhibitors [93]. Of interest, in HFD-fed C57BL/6 mice presenting a diabetic condition, endothelial progenitor cells (EPCs) were reduced in peripheral blood. However, irisin treatment slightly increased the proliferation and migration of EPCs. Since this effect is reduced by L-NAME, the direct regulation of eNOS was proposed as the underlying mechanism. The Western blotting approach showed the increased expression of phosphorylated eNOS and Akt, which were reduced after incubation with the PI3K inhibitor [135].

### 4.3. Effects of H_2_S and Irisin on Ion Channels

The vasorelaxing activity of H_2_S is attributed, in part, to the stimulation of ionic channels on vascular smooth muscle or the endothelium, such as through ATP-sensitive potassium channels (K_ATP_); voltage-gated potassium channels such as 7.4 (K_v_7.4); and TRPV4 channels [11,21,22,136].

The K_ATP_ channels are voltage-independent ligand-gated channels and represent one of the major pathways involved in vessel vasodilatation. These channels are constituted by four pore-forming subunits (Kir6.x) and four regulatory proteins (SURx), and the predominant combination in vascular smooth muscle cells is Kir6.1/SUR2B [137,138,139]. Sun et al. studied the aortic rings of normotensive and spontaneously hypertensive rats to evaluate the effect of H_2_S on K_ATP_ channels in vascular smooth muscle cells. In the aorta, the mesenteric artery, and the tail artery, the expression of the SUR2B and Kir6.1 proteins was lower in hypertensive rats compared with normotensive ones, but treatment with NaHS significantly upregulated the levels of these proteins [94]. Zhao et al. also showed the glibenclamide-dependent vasorelaxant response of H_2_S [12]. The Snyder group, in 2011, first explored the mechanism through which NaHS stimulates the K_ATP_ channels and found S-persulfidation at cysteine-43 on the Kir6.1 subunit [95]. Notably, Martelli et al. first furnished functional and electrophysiological evidence of the role of the Kv7.4 channels in the vasoactive effects of H_2_S. Besides the non-selective blockers of potassium channels, the Kv7 selective blockers XE 991 and linopirdine (each at 10 µM) abolished the vasoactive effects of NaHS. Moreover, since Rb^+^ is known as a “potassium-mimetic” cation, the role of potassium channels was also investigated, assessing Rb^+^-efflux. Even in this case, XE-991 (10 µM), linopiridine (10 µM), and glibenclamide (10 µM) abolished a NaHS-induced increase in Rb^+^-efflux. Finally, the authors observed that TEA, glibenclamide, linopirdine, and XE 991 markedly antagonized NaHS-induced hyperpolarization in human aortic smooth muscle cells [11].

The engagement of potassium channels located on the inner mitochondrial membrane (mitoK) has been suggested as the main mechanism through which H_2_S may promote myocardial protection [16,96,97,99,140]. In addition to the mitoK_ATP_ channels, recently, Testai and her colleagues demonstrated the S-persulfidation of mitoKv7.4 by erucin, an isothiocyanate H_2_S-donor present in *Brassicaceae* vegetables, as a target of anti-ischemic cardioprotection [15].

The activation of the potassium channels has also been viewed with irisin. Zhang et al. demonstrated that this myokine reduces blood pressure via hyperpolarization determined by the activation of K_ATP_ channels [100]. Demirel et al. demonstrated the concentration-dependent response of irisin (10^−9^–10^−6^ M) in precontracted aortic rings, and the vasorelaxant effect was observed both in endothelium-intact and -denuded preparations. Incubation with several potassium channel blockers selective for K_ATP_, as well as calcium-activated potassium channels and a non-selective one, showed the multitarget activity of this myokine [54]. To the best of our knowledge, no evidence has reported potassium-channel-mediated myocardial protection through this myokine.

In addition, TRPV4 channels regulate the entry of calcium into endothelial cells and vascular smooth muscle cells, thus activating vasodilation in vessels. Indeed, the increment of intracellular Ca^2+^ may trigger a NO-dependent vasodilatory effect [141,142]. In particular, the increment of Ca^2+^ cytosolic content enhances the formation of complex Ca^2+^-calmodulin, which activates eNOS, leading to an increment of NO production [143,144,145].

Recent studies affirmed that the H_2_S-dependent vasodilatory effect is partly mediated by TRPV4 channels. NaHS (1–10M) in phenylephrine-induced precontracted aortic rings promotes vasodilation, inhibited by pretreatment with a TRPV4 antagonist (GSK2193874A, 10 mM). This evidence also found that TRPV4 activation stimulated BK_Ca_ channels, finding hyperpolarization in the membrane. The major mechanism identified as responsible for TRPV4 activation is S-persulfidation, but further studies are necessary [98].

Likewise, in rat mesenteric arteries and rat mesenteric artery cells, irisin induces the release of calcium through the activation of TRPV4 channels. Indeed, Ye et al. first observed that Ca^2+^ influx is reduced by incubation with TRPV4 blockers, TRP (2-APB), and HC067047, confirming their main involvement [101].

### 4.4. Effects of H_2_S and Irisin on Nrf2

Nrf2 is a protein that regulates the expression of cytoprotective genes, exerting protective action against oxidative stress. In physiological conditions, Nrf2 belongs to an inactivated complex with the Keap1 protein and is activated when Keap1 is released. Under various stimuli, Keap1 may change its conformation and provoke dissociation from Nrf2, which translocates in the nucleus. By binding to DNA, Nrf2 regulates the expression of antioxidant genes [146,147]. Increasing studies demonstrate that H_2_S is an activator of the Nrf2 complex via the S-persulfidation of Keap1 on its Cys151, Cys226, and Cys613 residues [148,149]. Calvert et al. furnished evidence that H_2_S performs cardioprotective action—at least in part—via Nrf2 activation. In one study, H_2_S (100 µg/kg) intravenously injected into mice 24 h before MI reduced myocardial damage and improved the ejection fraction by 87% in comparison with control mice. An increment of Nrf2 expression and antioxidant proteins, i.e., heme-oxygenase (HO-1) and thioredoxin (Trx1), was observed in cardiac tissue after several hours following the H_2_S injection. To investigate whether this cardioprotective action was correlated with Nrf2 activation, the researchers repeated the experiments on Nrf2 KO mice, and in that case, H_2_S did not protect from myocardial injury [103]. Nrf2 KO mice were also studied by Shimizu et al., who observed no attenuation of heart failure-induced cardiac damage after treatment with Na_2_S (100g/kg). In contrast, in wild-type mice subjected to ischemic-induced heart failure, H_2_S improved Nrf2 signaling, left ventricular function, and reduced cardiac hypertrophy [104]. Similar results were obtained by Peake et al. for db/db diabetic mice after I/R injury [102].

The results of recent studies suggest that irisin also acts through the Nrf2 signaling pathway. The irisin-protective effect against ferroptosis and mitochondrial dysfunction in cardiomyocytes under hypoxic conditions is mediated by the activation of the Nrf2/HO-1 axis [60]. Of interest, in Zhu and colleagues’ work, streptozotocin-induced T2D mice were treated with irisin (0.5 µg/g/day) for 12 weeks, and an improvement in cardiac function was observed. In vitro studies have shown that irisin exerts antioxidant properties via the ERK1/Nrf2/HO-1 pathway, stimulating the translocation of Nrf2 and enhancing HO-1 expression. Conversely, siRNA Nrf2 abolishes the irisin-protective effect [106]. Luna-Ceron et al. elucidated the role of irisin in endothelial dysfunction, demonstrating that the antioxidant effect was mediated by the Akt/mTOR/Nrf2 pathway [150], and this myokine contributed to a reduction in oxLDL-induced vascular injuries [105]. Finally, Mazur-Bialy et al. demonstrated that irisin (0, 25, and 50 nM) protects LPS-activated RAW 264.7 macrophages via the Nrf2/HO-1 axis [107].

### 4.5. Effects of H_2_S and Irisin on Inflammasome

Nucleotide-binding oligomerization domain-like receptor protein 3 (NLRP3) is a well-known inflammasome complex implicated in CV diseases, including atherosclerosis, MI, diabetes, hypertension, and dilated cardiomyopathy [151]; for this reason, the inhibition of this inflammasome is considered a promising strategy for alleviating chronic diseases and reducing CV disease risk factors.

The H_2_S donor GYY4137 exerts a protective effect on diabetes-accelerated atherosclerotic endothelial cells and murine models [108], sepsis-induced cardiomyopathy patients, and mice [109,152]. Ni and colleagues demonstrated that H_2_S can block NLRP3-induced pyroptosis in murine models of I/R-induced kidney damage. This gasotransmitter alleviates aberrant changes in kidney tissue induced by treatment with PAG [110].

In parallel, Yue et al. conducted in vivo and in vitro studies and reported that irisin injection (2 g/kg/week) reduced NLRP3, caspase-1, and IL-1β levels in the cardiac tissue of C57/BL6 mice with cardiac hypertrophy. In vivo experiments confirmed the protective role of irisin since cardiomyocytes deriving from animals treated with Angiotensine II (1 ng/mL) plus irisin (100 ng/mL) showed reduced pyroptosis compared with control animals [111].

### 4.6. Effects of H_2_S and Irisin on AMPK/mTOR-Dependent Autophagy

Autophagy is a repair mechanism that regulates cellular homeostasis through the catabolism of proteins and cellular components or organelles. It is physiologically activated to guarantee cellular maintenance in cases of stress conditions, such as growth factor deprivation, hypoxia, and oxidative stress [153]. Autophagy is activated by AMPK phosphorylation and mTOR inhibition.

Yang et al. demonstrated, for the first time, that H_2_S increases AMPK activity in cardiac tissue in a murine diabetic model. STZ-induced diabetic rats were intraperitoneally treated with NaHS (100 µM) for 4–8 weeks, which resulted in an increment of autophagic vacuoles and the expression of p-AMPK/AMPK. In vitro experiments on H9c2 cardiomyocytes treated with glucose and GYY4137 (100 µM) revealed an improvement in cell viability with an increment of the p-AMPK/AMPK ratio and a reduction in p-mTOR/mTOR. Compound C, an AMPK inhibitor, abolished the cytoprotective effect of H_2_S [112]. AMPK/mTOR is involved in hepatic protection: NaHS improved autophagy in liver cells of HFD-fed mice leading to increased p-AMPK levels, reduced p-mTOR, and lowered triacylglycerol serum levels [154].

Like H_2_S activity, irisin activity is related to AMPK/mTOR activation. as demonstrated by in vitro and in vivo experiments. In particular, Deng et al.’s study revealed that irisin treatment (20 µM) prevents high-glucose-induced cardiomyocyte damage by activating the AMPK/mTOR pathway; conversely, its protective effect was abolished by the AMPK inhibitor [114].

### 4.7. Effects of H_2_S and Irisin on TGF-β1/Smad Pathway

Myocardial fibrosis is another crucial aspect of the pathological picture of cardiometabolic diseases. Transforming growth factor-β1 (TGF-β1) is considered a key player in the development of the fibrotic process and has emerged as a promising therapeutic target. Through the bond with specific Type I (TβR-I) and Type II (TβR-II) receptors, TGF-β1 determines the phosphorylation of Smad2 and Smad3 proteins, which regulate collagen synthesis and cardiac remodeling [155,156,157,158].

Sun and colleagues investigated the role of H_2_S in cardiac fibrosis, performing a 9-week treatment with NaHS (90 μmol/kg/day) on spontaneously hypertensive rats. Their findings showed that NaHS supplementation prevented myocardial collagen synthesis, decreasing collagen I, collagen III, TGF-β1, p-Smad2, and p-Smad3 protein expression. Moreover, in vitro studies on cardiac fibroblast cells treated with TGF-β1 and NaHS revealed a reduction in the phosphorylation of myocardial the TβR-I protein, Smad2, and Smad3; interestingly, this effect was abolished after incubation with a Smad inhibitor (SB-431542) [113]. Similar results were obtained by Meng et al., who demonstrated a reduction in cardiac fibrosis in spontaneously hypertensive rats daily injected with GYY4137 (25 and 30 mg/kg/day) for 1 month. At the end of this treatment, Picrosirius red staining showed a reduction in collagen fibers in myocardial tissue, and Western Blot images revealed a reduction in TGF-β1 and p-Smad2 expression in cardiac fibroblasts [115]. An antifibrotic effect was reported also by Yang et al. [116], Wang et al. [117], and Song et al. [118].

Regarding irisin, only one study has investigated the potential antifibrotic effects of irisin, suggesting that the TGF-β1/Smad pathway is an underlying mechanism. Therefore, Wu and colleagues recently demonstrated that irisin treatment (500 µg/kg/day) used on Ang II-infused mice reduced atrial fibrosis markers. In particular, a significant reduction in TGF-β1, p-Smad2, and p-Smad3 was observed in the atrial tissue of the treated mice [119].

### 4.8. Effects of H_2_S and Irisin on Glucose Uptake

Hyperglycemic conditions, insulin resistance, and free fatty acids are metabolic imbalances responsible for pathological conditions. These metabolic alterations determine an increment of oxidative stress that may provoke vascular inflammation, vasoconstriction, thrombosis, and atherogenesis. T2D patients have an increased risk of mortality compared with healthy individuals, and CV diseases are the primary T2D-associated mortality cause [159].

Insulin resistance is characterized by the impairment of several pathways [160]. Insulin may act on mitogen-activated protein kinase (MAPK)/extracellular signal-regulated kinase (ERK), which mediates signals to mainly regulate cell growth and proliferation; in insulin resistance, these mediators are impaired and may provoke metabolic diseases and CV-associated alterations [160,161].

In recent years, H_2_S has been identified as a modulator of glucose homeostasis, exploiting crucial pathways involved in blood glucose homeostasis [162]. H_2_S enhances glucose uptake, and in in vitro assays on myoblasts, an increment of phosphorylated AMPK, activated by the S-persulfidation mechanism, has been observed, along with the activation of the AMPK/p38 MAPK pathway [120]. Pichette and colleagues focused on the role of H_2_S in glucagon-like peptide-1 (GLP-1) expression, a hormone involved in glucose homeostasis and considered a therapeutical target for T2D and obesity treatment. GLUTag cells were treated with NaHS, GYY4137, or vehicle for 2 h, and an increase in GLP-1 secretion was observed. Since an increment of p-p38MAPK was revealed by Western blot images, the authors suggested that the activation of MAPK was the underlying mechanism. Thus, pretreatment with the p38 MAPK inhibitor blocked NaHS-induced GLP-1 secretion [121]. In contrast, Xu et al. described how exogenous H_2_S (400 µM) exerts cytoprotective effects on H9C2 cells in high-glucose conditions, reducing p-p38-MAPK and p-ERK1/2 content [122]. The role of H_2_S has even been demonstrated in in vivo studies: skeletal muscle conditional knockout CSE mice showed a reduction in GLUT4 expression and the downregulation of the insulin receptor substrate-1(IRS1)/PI3K/Akt signaling pathway [163]. Accordingly, GYY4137 supplementation reversed these effects, promoting glucose tolerance and insulin sensitivity. These studies provide evidence of H_2_S activity in maintaining insulin sensitivity and blood glucose homeostasis, impairing different pathways and mechanisms.

Preclinical studies have shown that irisin can increase glucose uptake and insulin sensibility. Marrano et al. reported interesting evidence that described how irisin effects are closely similar to GLP-1 effects: these hormones share various direct pancreatic effects, such as the induction of insulin biosynthesis and glucose-dependent insulin secretion, as well as the improvement of β-cell proliferation [164]. Recent studies have supposed that both GLP-1 and irisin are synthesized by pancreatic cells and that irisin acts on αV integrin receptors [47,164]. Lee et al. studied irisin’s hypoglycemic effect on rat L6 myoblasts: the peak of uptake was observed at an irisin concentration of 62 ng/mL [126]. Western blot images showed an increment of p-AMPK content, and pretreatment with compound C inhibited the irisin-induced increment of glucose uptake. The authors focused on the possible underlying mechanism, hypothesizing that it may be related to the role of p38-MAPK. Treatment with irisin determined an increment of p-p38-MAPK content in L6 cells, and pretreatment with a specific blocker (SB202190) abolished the glucose uptake. Finally, the silencer of p38-MAPK genes reduced glucose absorption [126].

## 5. Conclusions

The gasotransmitter H_2_S and the myokine irisin are two endogenous mediators involved in the homeostasis of organisms and implicated in the pathogenesis of CV and metabolic disorders. In this regard, based on the evidence reported in this work, both H_2_S and irisin could represent useful diagnostic and prognostic tools for CV diseases, even for metabolic disorder conditions. In particular, as reported above, lower levels of H_2_S have been found in patients affected by CV morbidities, and the levels of irisin seem to be correlated with the severity of coronary artery diseases. Therefore, pharmacological intervention with molecules capable of releasing H_2_S and/or the administration of irisin itself could represent a promising approach to the management of CV diseases. However, some issues need to be taken into consideration. H_2_S donors should release this gasotransmitter with slow and constant kinetics capable of providing a long-lasting supply over time while in the meantime avoiding the onset of toxic effects related to high concentrations of H_2_S. Since 2010, numerous synthetic and naturally derived H_2_S donors have been described [165]. In this regard, some vegetables have been shown to contain secondary metabolites endowed with H_2_S-releasing properties, such as those in the *Brassicaceae* family [166] and the *Alliaceae* family [167] and, more recently, several species of mushrooms [168] and marine organisms [169].

On the other hand, the direct administration of irisin also presents a series of pharmacokinetic issues.

Recent studies have brought to light the effects of the parenteral administration of recombinant irisin on animal models of CV damage; nevertheless, given the obvious problems inherent to the parenteral administration of this myokine, great attention has been focused on possible molecules that can be administered enterally and that are capable of modulating the irisin pathway, allowing for its endogenous release. The pharmacodynamic target of irisin is a further aspect to be clarified, which hinders the pharmacological approach with molecules capable of acting on the same molecular target.

From this perspective, several drugs currently on the market, such as simvastatin, ezetimibe, fenofibrate, metformin, and sitagliptin, have been described for their ability to positively modulate the irisin pathway by increasing its endogenous production. Even dietary components, such as polyphenols and flavonoids, appear to be effective in increasing the endogenous release of irisin [6].

Finally, the possibility of modulating one of these systems is even more interesting considering that H_2_S and irisin share an impressive amount of intracellular pathways. This work, for the first time, describes possible crosstalk between H_2_S and irisin, allowing us to “wake up” to this possible interconnection and the real benefit it could have from a diagnostic/prognostic perspective. The relevance of this crosstalk between H_2_S and irisin also spreads in the pharmacological direction. The pharmacological modulation of one could influence the other and vice versa, generating synergistic effects.

## Figures and Tables

**Figure 1 antioxidants-13-00543-f001:**
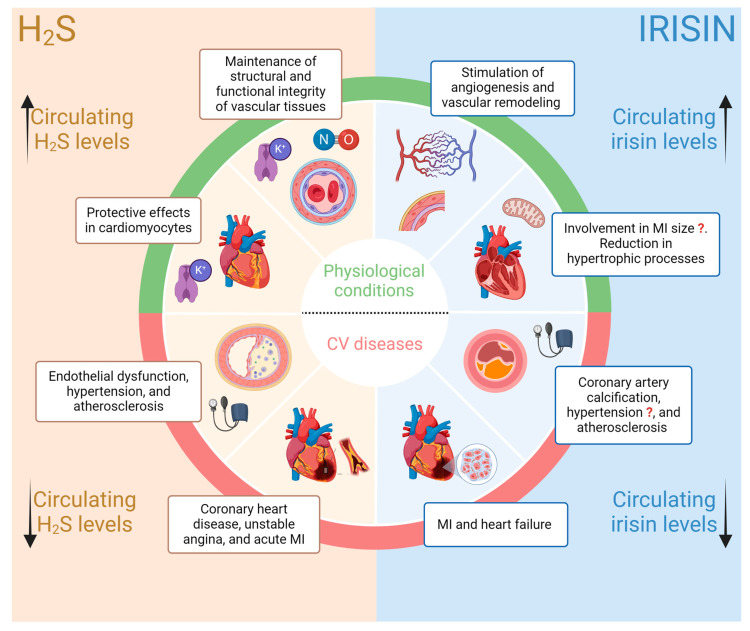
Graphic representation of the effects of H_2_S and irisin on physiological conditions and CV diseases and changes in plasma concentrations of H_2_S and irisin in CV diseases that could kick off their use as diagnostic/prognostic markers. The up arrow means increase, the down arrow means decrease, and ? means that the experimental evidence does not yet agree on increase or reduction but underlines a change.

**Table 1 antioxidants-13-00543-t001:** Secretion, mechanism, activity, and variation in circulation levels of H_2_S and irisin. ↓: decreased; ↓↑: decreased or increased in different studies.

	H_2_S	Irisin
Secretion	- Non-enzymatic secretion: reduction of elemental sulfur to H_2_S with the oxidation of six molecules of glucose [9]. - Enzymatic secretion: CSE. CBS and 3-MST secrete H_2_S from L-cysteine in central nervous system and CV system [9].	- Proteolytic secretion: A proteolytic cleavage from fibronectin type III domain-containing protein 5 (FNDC5) in heart, brain, skeletal muscle, liver, kidneys, lung, spleen, skin, and adipose tissue [6,46].
Mechanism	Post-translational modification (S-persulfidation) [2].	Interaction with αV/β5 integrin receptors [47].
Activity	Maintenance of the structural and functional integrity of vascular tissue; cardioprotection; containment of inflammation; reduction in oxidative stress.	Stimulation of browning process;promotion of neovascularization and angiogenesis; vasorelaxant effect; anti-atherosclerotic activity; reduction in oxidative stress.
Circulating levels variation	↓ vascular remodeling [25] ↓ atherosclerosis condition [25] ↓ cardiac dysfunction [28] ↓ coronary heart disease [36] ↓ obesity condition [25,30] ↓ diabetic condition [31,32,33,34,35] ↓ acute inflammatory conditions [38]	↓ coronary artery disease [63,64,65,66] ↓↑ acute MI and heart failure [67,68,69] ↓ acute ischemic stroke [76] ↓ atherosclerosis condition [70] ↓ obesity condition [46,71,72] ↓ diabetic condition [46,71,72] ↓ cerebrovascular diseases [75] ↓↑ hypertension condition [77,78,79,80]

**Table 2 antioxidants-13-00543-t002:** In this table, in vitro and in vivo H_2_S and irisin studies are reported. In the modulation of common or different pathways, H_2_S and irisin final outcomes are so similar that an interesting form of crosstalk between these molecules might be assumed.

PATHWAY	H_2_S	IRISIN
SIRT1/AMPK/PGC1α	Cardioprotection observed in H9c2 cardiomyocytes treated with NaHS (25–100 µmol/L) [81]. Cardioprotection against I/R injury and reduction in infarct size in heart perfuse with NaHS (10 µmol/L) [82]. Reduction in myocardial hypertrophic markers and restored cardiac function observed in WT and CSE-KO mice injected with isoprenaline and NaHS (56 µmol/kg/day) for 4 weeks [83]. Regulation of mitochondrial biogenesis in CSE TG^+^, CSE-KO, and AMPK-KO mice treated with SG1002 (20 mg/kg/day) [84].	Regulation of mitochondrial biogenesis in transgenic mice injected with adenoviral vectors [45].
eNOS/NO	Protection against vascular lesions in HUVEC cells treated with H_2_S donors [85] Negative inotropic effect observed in heart perfused with NaHS (10^−11^–10^−8^M) [86]. Decrement of blood pressure in rats treated with L-NAME and NaHS for 5 weeks [87]. Decrement of blood pressure in rats treated with L-NAME and NaHS (20 µmol/kg per day) for 6 weeks [88]. Prevention of L-NAME-induced hypertension and attenuation of cardiac dysfunction and remodeling observed in L-NAME-induced hypertensive rats treated with NaHS (56 μmol/kg/day) for 5 weeks [89]. Cardioprotection against I/R injury in CSE-KO and eNOS KO mice injected with Na_2_S (100 μg/kg) 5 min before reperfusion [90].	Vasorelaxant effect and improvement of endothelial dysfunctions in aortic rings of obese mice treated with irisin (0.5 µg/g/day) for 8 weeks [91]. Prevention of phenylephrine-induced vasoconstriction in mesenteric arterial segments preincubated with irisin (3000 ng/mL) [92]. Anti-hypertensive effect on SHR rats treated with irisin (0.1, 1, and 10 µg/kg) [92]. Vasorelaxant effect on and improved endothelial function in Streptozotocin-induced diabetic ApoE−/− mice treated with irisin (2 µg/mouse) [93].
Ionic channels K_ATP,_ K_v_7.4, TRPV4	K_ATP_-mediated vasorelaxant effect in aortic rings of Wistar and SHR rats [94]. K_ATP_-mediated vasorelaxant effect in rat aortic rings incubated with H_2_S (180 µM) [12]. K_ATP_-mediated vasorelaxant effect on rat mesenteric arteries with H_2_S (100 µmol/L) [95]. K_ATP_-mediated vasorelaxant effect on smooth muscle cells treated with H_2_S or NaH_2_ (0.01 µM–10 Mm) [21]. K_ATP_-mediated protection against myocardial I/R injury in Wistar rat hearts [16,96,97]. K_ATP_- and Kv7-mediated vasorelaxant effect on rat aortic rings incubated with NaHS (E_max_ 1 mM) [11]. TRPV4-mediated vasorelaxant effect on rat aortic rings treated with NaHS (1–10 M) [98]. MitoBKCa-mediated cytoprotection [99].	K ATP-mediated vasorelaxant effect on mesenteric rat arteries [100]. Kv-, K_ATP_-, and BKCa-mediated vasorelaxant effect on rat aortic rings treated with irisin (10^−9^–10^−6^ M) [54]. TRPV4-mediated vasorelaxant effect on rat mesenteric arteries and rat mesenteric artery cells treated with irisin (100 nM) [101].
Nrf2	Protection against myocardial injury after I/R in db/db diabetic mice (0.1 mg/kg/day) [102]. Protection against myocardial injury in Nrf2-KO and control mice injected with H_2_S (100 µg/kg) [103]. Protection against heart-failure-induced cardiac damage in Nfr2-KO mice treated with Na2S (100 g/kg) [104].	Protection against ferroptosis and mitochondrial dysfunction [60]. Antioxidant effect observed in HUVEC cells [105]. Improvement of cardiac functions in streptozotocin-induced T2DM mice treated with irisin (0.5 µg/g/day) for 12 weeks [106]. Protection of LPS-activated RAW 264.7 macrophage treated with irisin (0, 25, and 50 nM) [107].
Inflammasome NLRP3	Anti-atherosclerotic effect on HUVEC cells treated with GYY4137 (100 μmol/L) [108] Reduction in cardiac inflammatory response in sepsis studied in wild-type and Nlrp3^−/−^ mice treated with GYY4137 (50 mg/kg) [109]. Inhibition of pyroptosis process in mice treated with NaHS (500 μg/kg/day) and HK2 cells pretreated with NaHS (100 μM) [110].	Anti-inflammatory effect on cardiac tissue of C57BL/6 mice presenting cardiac hypertrophy and treated with irisin (2 g/kg/week) [111]. Inhibition of pyroptosis process observed in mice cardiomyocytes treated with irisin (100 ng/mL) [111].
AMPK/mTOR	Improvement of cell viability in H9C2 cardiomyocytes treated with glucose and GYY4137 (100 µM) [112]. Increment of autophagic process in STZ-induced diabetic rats injected with NaHS (100 µM) for 4–8 weeks [112]. Increased autophagic process in liver cells treated with NaHS [113].	Cardioprotective effect on high-glucose-damaged cardiomyocytes treated with irisin (20 µM) [114].
TGF-β1/Smad	Reduction in collagen I, collagen III, TGF-β1, p-Smad2, and p-Smad3 protein expression in SHR rats treated with NaHS (90 μmol/kg/day) for 9 weeks [113] Reduction in collagen fibers, TGF-β1, and p-Smad2 expression in SHR rats injected with GYY4137 (25–30 mg/kg/day) for 1 month [115]. Reduction in profibrotic markers and collagen fiber deposition in STZ-induced diabetic rats treated with NaHS (56 µmol/kg/day) [116]. Reduction in aorta calcification in diabetic rats treated with NaHS (50 μmol/kg/day) for 8 weeks [117]. Alleviated hypothyroidism-induced myocardial fibrosis in rats treated with NaHS (56 µmol/kg/d) for 4 weeks [118].	Reduction in atrial fibrosis markers and collagen in Ang II-infused mice treated with irisin (500 µg/kg/day) [119].
Glucose homeostasis AMPK/p38 MAPK	Increment of glucose uptake in NaHS (50 µmol/kg/day)-treated chicks [120]. Rise in GLP1 secretion observed in GLUTag cells treated with NaHS (10 Mm) or GYY4137 (100 µM) [121]. Cytoprotective effect on H9c2 cells exposed to high glucose and treated with NaHS (400 µM) [122].	Increased glucose uptake in irisin-overexpressed C2C12 myoblasts [123]. Improved glucose homeostasis and increased hepatic glycogenesis synthesis in human-insulin-resistant HepG2 cells treated with irisin (20 nM) [124]. Increased GLUT4 expression and glucose uptake in irisin-treated C2C12 cells [125]. Increased glucose uptake in rat L6 myoblasts treated with irisin (peak at 62 ng/mL) [126]. Improved glucose metabolism in diabetic rats injected daily with irisin (0.5 μg/g) for 10 days [125]. Increment of muscle glycogen stored in mice treated with irisin (0.1 mg/kg/4 times a week) for 4 weeks [123]. Improved insulin sensitivity in the STZ/HFD mice treated with irisin (1.44 nmol/day) for 14 days [124].

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
