# Peer review of "Hydrogen Sulfide and Irisin, Potential Allies in Ensuring Cardiovascular Health"

_antioxidants, 2024, doi:10.3390/antiox13050543_

Round 1

Reviewer 1 Report

The aim of this review is to discuss the relationship between the gasotransmitter hydrogen sulfide and irisin. In particular, parallel factors regulating the secretion of both mediators as well as their common and divergent activities, especially in the cardiovascular system and metabolic effects, were discussed. The effect of H2S on irisin production reported in one experimental study is also covered.

The topic is of interest, however, I have also some concerns to be addressed.

1)     Line 70, “hydrosulfhydryl groups” should be corrected to: “sulfhydryl groups”.

2)     Line 154, serum H2S level is very low and few studies addressed the relationship between true serum H2S (rather than total amount of reactive sulfur species such as sulfane sulfur) and human diseases have been performed. Thus, it is highly questionable if serum H2S could be clinically relevant marker.

3)     Line 165: the level of free H2S is within low nanomolar rather than micromolar range.

4)     Line 184, protein glycosylation is the posttranslational, not translational, modification.

5)     Lines 213-215: stimulatory effects of irisin on smooth muscle cell proliferation and plaque growth are contradictory to its protective effects on the cardiovascular system.

6)     Lines 220221: “the size of MI” should rather be revised to: “the infarct size”.

7)     Line 230: with which cv risk factors irisin is positively correlated?

8)     Line 242: the specific adverse cv events should be listed.

9)     Line 245, “atherogenic index” should be explained.

10) Lines 249/250: what alterations of the lipid metabolism are mentioned?

11) It would be reasonable to include the table presenting similarities and differences between irisin and H2S regarding factors which regulate their secretion and their activities.

12) The only original study about the effect of H2S on irisin (ref. 8) should be discussed in more details (section 4)

13) Line 342: “endothelium dysfunction” should be corrected to “endothelial dysfunction”

14) Title of section 4.3: “ionic channels” should be corrected to: “ion channels”

The aim of this review is to discuss the relationship between the gasotransmitter hydrogen sulfide and irisin. In particular, parallel factors regulating the secretion of both mediators as well as their common and divergent activities, especially in the cardiovascular system and metabolic effects, were discussed. The effect of H2S on irisin production reported in one experimental study is also covered.

The topic is of interest, however, I have also some concerns to be addressed.

1)     Line 70, “hydrosulfhydryl groups” should be corrected to: “sulfhydryl groups”.

2)     Line 154, serum H2S level is very low and few studies addressed the relationship between true serum H2S (rather than total amount of reactive sulfur species such as sulfane sulfur) and human diseases have been performed. Thus, it is highly questionable if serum H2S could be clinically relevant marker.

3)     Line 165: the level of free H2S is within low nanomolar rather than micromolar range.

4)     Line 184, protein glycosylation is the posttranslational, not translational, modification.

5)     Lines 213-215: stimulatory effects of irisin on smooth muscle cell proliferation and plaque growth are contradictory to its protective effects on the cardiovascular system.

6)     Lines 220221: “the size of MI” should rather be revised to: “the infarct size”.

7)     Line 230: with which cv risk factors irisin is positively correlated?

8)     Line 242: the specific adverse cv events should be listed.

9)     Line 245, “atherogenic index” should be explained.

10) Lines 249/250: what alterations of the lipid metabolism are mentioned?

11) It would be reasonable to include the table presenting similarities and differences between irisin and H2S regarding factors which regulate their secretion and their activities.

12) The only original study about the effect of H2S on irisin (ref. 8) should be discussed in more details (section 4)

13) Line 342: “endothelium dysfunction” should be corrected to “endothelial dysfunction”

14) Title of section 4.3: “ionic channels” should be corrected to: “ion channels”

Author Response

Reviewer 1

The aim of this review is to discuss the relationship between the gasotransmitter hydrogen sulfide and irisin. In particular, parallel factors regulating the secretion of both mediators as well as their common and divergent activities, especially in the cardiovascular system and metabolic effects, were discussed. The effect of H2S on irisin production reported in one experimental study is also covered.

The topic is of interest, however, I have also some concerns to be addressed.

  • Line 70, “hydrosulfhydryl groups” should be corrected to: “sulfhydryl groups”.

Reply: Thank you so much, we fixed it.

  • Line 154, serum H2S level is very low and few studies addressed the relationship between true serum H2S (rather than total amount of reactive sulfur species such as sulfane sulfur) and human diseases have been performed. Thus, it is highly questionable if serum H2S could be clinically relevant marker.

Reply: Thanks for making this point. We believe it is important that, on one hand the key role of H2S and its endogenous production in CV homeostasis emerges but on the other one that the difficulties to consider circulating H2S levels as a reliable diagnostic marker of CV pathologies are deeply underlined. This aspect has already been highlighted by line 181 to line 195. Considering your timely and appropriate intervention, we have therefore modified the paragraph indicated by you, further clarifying just the intriguing possibility of taking into consideration the circulating levels of H2S, with all the related problems, for the identification of CV pathologies and their severity.

  • Line 165: the level of free H2S is within low nanomolar rather than micromolar range.

Reply: We made the suggested change.

  • Line 184, protein glycosylation is the posttranslational, not translational, modification.

Reply: Thank you so much, we fixed it.

  • Lines 213-215: stimulatory effects of irisin on smooth muscle cell proliferation and plaque growth are contradictory to its protective effects on the cardiovascular system.

Reply: Thanks for pointed out this aspect. This part has been modified for better understanding.

The study was reported in the first part of the paragraph with the aim of introducing how irisin can be involved in morpho/functional changes through remodeling processes. In fact, this experimental analysis seems to fight with the positive effects of irisin on the cardiovascular district. However, it should be underlined that the evidence of the effects of irisin and its systemic levels in conditions of hypertension are conflicting and still not perfectly clear, as explained in the last part of this paragraph.

  • Lines 220-221: “the size of MI” should rather be revised to: “the infarct size”.

Reply: We made the change.

  • Line 230: with which cv risk factors irisin is positively correlated?

Reply: Thank you so much for pointed out this issue. In this context, our mind was to indicate a clear link between circulating irisin levels, cardiovascular pathologies such as MI, hypertension, endothelial dysfunction, and cardiovascular risk factors such as dyslipidemia, diabetes, and obesity.

To this end, the sentence has been slightly modified to better clarify this aspect.

The positive or negative correlation indicative of the increase or reduction of circulating irisin levels in relation to cardiovascular outcomes is indicated at the end of each experimental evidence reported in this paragraph.

  • Line 242: the specific adverse cv events should be listed.

Reply: Thank you for the suggestion. We improved this point.

  • Line 245, “atherogenic index” should be explained.

Reply: We have better explained what it is and how it is calculated.

  • Lines 249/250: what alterations of the lipid metabolism are mentioned?

Reply: The mentioned lipid metabolism was better explained pointing out specific parameters evaluated.

  • It would be reasonable to include the table presenting similarities and differences between irisin and H2S regarding factors which regulate their secretion and their activities.

Reply: We inserted a concise table with these features in comparison.

  • The only original study about the effect of H2S on irisin (ref. 8) should be discussed in more details (section 4)

Reply: We have better explained what the study is about and the main findings

  • Line 342: “endothelium dysfunction” should be corrected to “endothelial dysfunction”

Reply: We fixed it.

  • Title of section 4.3: “ionic channels” should be corrected to: “ion channels”

Reply: Thank you so much, we did the suggested change.

Reviewer 2 Report

The paper explores the potential synergistic effects of hydrogen sulfide (H2S) and irisin in cardiovascular health. It discusses how these molecules, through common or different pathways, exhibit similar outcomes, suggesting a crosstalk between them. The review aims to investigate the roles of H2S and irisin in cardiovascular diseases, highlighting their effects on common molecular pathways. Experimental evidence supports the notion that irisin plays a crucial role in metabolic homeostasis and cardiometabolic regulation, while H2S acts as a gasotransmitter with pleiotropic functions. The review emphasizes the importance of understanding the interplay between these molecules for potential therapeutic interventions in cardiovascular diseases. Specific comments:

1.          The introduction could benefit from a more detailed explanation of the physiological roles of hydrogen sulfide (H2S) and irisin. This would help readers understand the significance of the study.

2.          The discussion section could delve deeper into the implications of the findings for cardiovascular health and metabolic regulation.

3.          Could you elaborate on the mechanisms by which H2S contributes to vasodilation?

4.          How does H2S enhance nitric oxide (NO) signaling?

5.          Could you provide more evidence to support the claim that H2S has cardioprotective properties?

6.          The paper mentions that H2S affects glucose metabolism and insulin sensitivity. Could you discuss these effects in more detail?

7.          Could you elaborate on the role of irisin in endothelial dysfunction?

8.          The conclusion could summarize the main findings more succinctly and highlight the potential applications of the research.

The paper explores the potential synergistic effects of hydrogen sulfide (H2S) and irisin in cardiovascular health. It discusses how these molecules, through common or different pathways, exhibit similar outcomes, suggesting a crosstalk between them. The review aims to investigate the roles of H2S and irisin in cardiovascular diseases, highlighting their effects on common molecular pathways. Experimental evidence supports the notion that irisin plays a crucial role in metabolic homeostasis and cardiometabolic regulation, while H2S acts as a gasotransmitter with pleiotropic functions. The review emphasizes the importance of understanding the interplay between these molecules for potential therapeutic interventions in cardiovascular diseases. Specific comments:

1.          The introduction could benefit from a more detailed explanation of the physiological roles of hydrogen sulfide (H2S) and irisin. This would help readers understand the significance of the study.

2.          The discussion section could delve deeper into the implications of the findings for cardiovascular health and metabolic regulation.

3.          Could you elaborate on the mechanisms by which H2S contributes to vasodilation?

4.          How does H2S enhance nitric oxide (NO) signaling?

5.          Could you provide more evidence to support the claim that H2S has cardioprotective properties?

6.          The paper mentions that H2S affects glucose metabolism and insulin sensitivity. Could you discuss these effects in more detail?

7.          Could you elaborate on the role of irisin in endothelial dysfunction?

8.          The conclusion could summarize the main findings more succinctly and highlight the potential applications of the research.

Author Response

Reviewer 2

The paper explores the potential synergistic effects of hydrogen sulfide (H2S) and irisin in cardiovascular health. It discusses how these molecules, through common or different pathways, exhibit similar outcomes, suggesting a crosstalk between them. The review aims to investigate the roles of H2S and irisin in cardiovascular diseases, highlighting their effects on common molecular pathways. Experimental evidence supports the notion that irisin plays a crucial role in metabolic homeostasis and cardiometabolic regulation, while H2S acts as a gasotransmitter with pleiotropic functions. The review emphasizes the importance of understanding the interplay between these molecules for potential therapeutic interventions in cardiovascular diseases.

Specific comments:

  1. The introduction could benefit from a more detailed explanation of the physiological roles of hydrogen sulfide (H2S) and irisin. This would help readers understand the significance of the study.

Reply: Thank you for the suggestion. We inserted a more detailed explanation of the physiological role of hydrogen sulfide and irisin in the cardiovascular system, reporting, where possible, the most relevant targets.

  1. The discussion section could delve deeper into the implications of the findings for cardiovascular health and metabolic regulation.

Reply: Thank you so much for pointing out this aspect. Considering the additions made in the different paragraphs, the table suggested by your colleague and the extension of the discussion about the findings in the conclusions, we think we have met your suggestion.

  1. Could you elaborate on the mechanisms by which H2S contributes to vasodilation?

Reply: we inserted a sentence in which we indicated the main mechanisms by which the gasotransmitter may contribute to vasodilation.

  1. How does H2S enhance nitric oxide (NO) signaling?

Reply: several authors, including us, investigated on the cross-talk between H2S and NO. Actually, the nature of the interaction between the gasotransmitters is complex and not yet completely clear. This review is focused on other aspects, however we tried to introduce it, but without distorting the theme of the review. We hope it can be satisfying.

  1. Could you provide more evidence to support the claim that H2S has cardioprotective properties?

Reply: Thank you for the suggestion. We speculated on the intracellular and intramitochondrial events correlated to the opening of mitochondrial potassium channels and cardioprotection. Moreover, we inserted another sentence in which reported other possible mechanisms engaged by H2S.

  1. The paper mentions that H2S affects glucose metabolism and insulin sensitivity. Could you discuss these effects in more detail?

Reply: We inserted more informations about H2S mechanism of action.

  1. Could you elaborate on the role of irisin in endothelial dysfunction?

Reply: Thank you for the comment, unfortunatly at the moment the role of irisin in endothelial damage remains unclear; however we inserted a sentence to emphasize the importance of this role;

  1. The conclusion could summarize the main findings more succinctly and highlight the potential applications of the research.

Reply: We agree on the need to better emphasize the evidence found. The key aspects have been emphasized following your suggestion.